# Reconstruction of a Neglected, Extensor Hallucis Longus Tendon Rupture Using Interposed Scar Tissue: A Case Report and Literature Review

**DOI:** 10.3390/ijerph182212157

**Published:** 2021-11-19

**Authors:** Woo-Jong Kim, Ki-Jin Jung, Hyein Ahn, Eui-Dong Yeo, Hong-Seop Lee, Sung-Hun Won, Dhong-Won Lee, Jae-Young Ji, Sung-Joon Yoon, Yong-Cheol Hong

**Affiliations:** 1Department of Orthopaedic Surgery, Soonchunhyang University Hospital Cheonan, 31, Suncheonhyang 6-gil, Dongam-gu, Cheonan 31151, Korea; kwj9383@hanmail.net (W.-J.K.); c89546@schmc.ac.kr (K.-J.J.); yunsj0103@naver.com (S.-J.Y.); 2Department of Pathology, Soonchunhyang University Hospital Cheonan, 31, Suncheonhyang 6-gil, Dongam-gu, Cheonan 31151, Korea; hyein.ahn@schmc.ac.kr; 3Department of Orthopaedic Surgery, Veterans Health Service Medical Center, Seoul 05368, Korea; angel_doctor@naver.com; 4Department of Foot and Ankle Surgery, Nowon Eulji Medical Center, Eulji University, 68, Hangeulbiseok-ro, Nowon-gu, Seoul 01830, Korea; sup4036@naver.com; 5Department of Orthopaedic Surgery, Soonchunhyang University Hospital Seoul, 59, Daesagwan-ro, Yongsan-gu, Seoul 04401, Korea; orthowon@schmc.ac.kr; 6Department of Orthopaedic Surgery, Konkuk University Medical Center, 120-1, Neungdong-ro, Gwangjin-gu, Seoul 05030, Korea; bestal@naver.com; 7Department of Anesthesiology and Pain Medicine, Soonchunhyang University Hospital Cheonan, 31, Suncheonhyang 6-gil, Dongam-gu, Cheonan 31151, Korea; phmjjy@naver.com

**Keywords:** foot, extensor hallucis longus, reconstruction, scar tissue

## Abstract

Injury of the extensor hallucis longus (EHL) tendon is relatively rare, but surgical repair is necessary to prevent deformity and gait disturbance. Primary suturing is possible if the condition is acute, but not when it is chronic. The scar tissue between the ruptured ends is a proliferative tissue composed of fibroblasts and collagen fibers. Given the histological similarity to normal tendons, several studies have reported tendon reconstruction using scar tissue. Here, we report a reconstruction of a neglected EHL rupture using interposed scar tissue. A 54-year-old female visited our clinic with a weak extension of a big toe. She had dropped a knife on her foot a month prior, but did not go to hospital. The wound had healed, but she noted dysfunctional extension of the toe and increasing pain. Magnetic resonance imaging (MRI) revealed that EHL continuity was lost and that the proximal tendon stump was displaced toward the midfoot. Scar tissue running in the direction of the original ligament was observed between the ruptured ends. In the surgical field, the scar tissue formed a shape similar to the extensor tendon. Therefore, we performed tendon reconstruction using the interposed scar tissue. For the first 2 postoperative weeks, the ankle and foot were immobilized to protect the repair. Six weeks after surgery, the patient commenced full weight-bearing. At the 3-month follow-up, active extension of the hallux was possible, with a full range of motion. The patient did not feel any discomfort during daily life. Postoperative MRI performed at 1 year revealed that the reconstructed EHL exhibited homogeneously low signal intensity, and was continuous. The AOFAS Hallux Metatarsophalangeal-Interphalangeal scale improved from 57 to 90 points and the FAAM scores improved from 74% to 95% (the Activities of Daily Living subscale) and from 64% to 94% (the Sports subscale). Scar tissue reconstruction is as effective as tendon autografting or allografting, eliminates the risk of donor site morbidity and infection, and requires only a small incision and a short operative time.

## 1. Introduction

Injury of the extensor hallucis longus (EHL) tendon is relatively rare, which is most commonly caused by a laceration when a sharp object is dropped on the dorsum of the foot [1,2]. In addition, this injury is associated with diabetes, rheumatoid arthritis, local steroid injections, and iatrogenic error during ankle arthroscopy [3,4]. Surgical treatment of the rupture is necessary to eliminate the risk of apical deformity and gait disturbance [5]. Primary suturing is possible if the condition is acute, but if the rupture is chronic, the gap between the tear edges widens due to tendon contraction, precluding end-to-end repair. In these cases, tendon transfer, autografting or allografting may be performed. However, these are associated with donor site morbidity and possible disease transmission [6].

The scar tissue in the gap between the rupture ends is a proliferative granulation tissue composed of fibroblasts and collagen fibers [7,8]. Given its histological similarity to normal tendons, several studies have reported successful reconstructions of chronic Achilles tendon ruptures using scar tissue. In addition, Yeo et al. reported favorable outcomes after chronic rupture of the extensor digitorum longus (EDL) tendon [9,10,11]. These procedures afford the advantages of a short operation time and no risk of donor morbidity or infection [11]. However, reconstruction of a chronic EHL rupture using scar tissue has not yet been reported. Here, we report the successful reconstruction of a neglected EHL rupture.

## 2. Case Presentation

This case report was approved by the Institutional Review Board of Soonchunhyang University Hospital (IRB No. 2021–07–754). The patient gave written informed consent for the publication of the report and the accompanying images.

### 2.1. Preoperative Evaluation

A 54-year-old female visited our clinic complaining of progressive dorsal foot pain in the flexion and weak extension of a big toe. She had dropped a knife on the foot dorsum proximal to the first metatarsophalangeal (MP) joint about 1 month prior, but did not go to the hospital, rather dressing the wound herself at home. The wound healed 2 weeks later, but she noted dysfunctional extension of the toe and increasing pain. Otherwise, she was healthy, with no sign of rheumatoid arthritis, no history of steroid use, and no neurological disorder.

Physical examination revealed a transverse scar 1 cm long that was 2 cm proximal to the first MP joint. A dimple was palpable over the scar. She complained of focal tenderness and mild swelling around the scar. The big toe was plantar-flexed, and the extension strength of the interphalangeal joint was Grade 1 (Figure 1). Passive extension was possible, with a full range of motion. The sensory function of the whole foot was normal. There was no biomechanical dysfunction of other ligaments except for EHL. In addition, there was no deformity in other joints of the foot. The American Orthopedic Foot and Ankle Society (AOFAS) Hallux MP-Interphalangeal scale score was 57. The Foot and Ankle Ability Measure (FAAM) scores on the Activities of Daily Living (ADL) and Sports subscales were 74% and 64%, respectively.

Plain radiographs of the weight-bearing foot revealed no specific findings. T2-weighted sagittal magnetic resonance imaging (MRI) revealed that EHL continuity was lost in the proximal region of the first metatarsal head, and that the proximal tendon stump was displaced toward the midfoot. In addition, the scar tissue of diffuse and a highly heterogeneous signal intensity was apparent in the original position of the ligament, running between the ruptured ends of the EHL (Figure 2). Otherwise, no other abnormal findings of ligaments or muscles were observed on MRI. Therefore, we diagnosed a neglected chronic EHL rupture, and planned surgery.

### 2.2. Surgical Procedure

Under general anesthesia, the patient was placed supine and a thigh tourniquet was applied to the affected side. A dorsal zigzag incision was made along the EHL from the scar to both ruptured ends. After dissecting the soft tissue, the rupture site and the scar tissue were exposed (Figure 3). The surgical field revealed no sign of infection. The distance between the ruptured ends was 5 cm, and was filled with a scar tissue similar in shape to the extensor tendon. When the scar tissue was pulled, the appropriate tension was maintained and the interphalangeal joint was extended. As the scar tissue could thus withstand the tension required for thumb extension, we decided to reconstruct the ligament using the scar tissue. After resection of about 0.7 cm of the proximal part of the scar (Figure 4a), the tissue was sutured to the proximal tendon stump (using the modified Kessler technique). We placed the epitendinous sutures of a no. 3–0 absorbable multifilament (Figure 4b). The ankle joint was in the neutral position and the MP joint in 5° of extension during suturing. When the tendon was pulled after suturing, the appropriate tension was maintained and the big toe was extended (Figure 5). Then, the wound was sutured layer-by-layer.

### 2.3. Postoperative Details

For the first 2 postoperative weeks, the ankle and MP joint were immobilized in the neutral position using a non-weight-bearing below-knee cast to protect the wound and the repaired tendon. Then, passive extension of the MP joint (via manual manipulation) was commenced to prevent adhesions. Gradual weight-bearing (in boots) and active hallux extension commenced 4 weeks postoperatively. Six weeks after surgery, the patient removed her brace and commenced full weight-bearing.

A specimen of the resected scar tissue was fixed in formalin, paraffin-embedded, and stained with hematoxylin-and-eosin. Under the microscope, bundles of collagen fibers running along the tendon, fibroblasts, and proliferative fibrovascular tissue were apparent (Figure 6). The collagen fibers were thinner than the intact tendons. However, we noted no changes in degenerative tissue, such as fatty infiltration or mucoid degeneration.

At the 3-month follow-up, active extension of the hallux was possible over the full range of motion. The patient was asymptomatic, and reported no discomfort in daily life. At 6 months after surgery, the extension strength was very close the intact side, and tendon contraction was evident under the skin (Figure 7). Postoperative MRI performed at 1 year revealed that the reconstructed EHL was of homogeneous low-level signal intensity, and was continuous (Figure 8). The AOFAS Hallux MP-Interphalangeal scale improved from 57 to 90 and the FAAM scores ranged from 74% to 95% (ADL subscale) and 64% to 94% (Sports subscale).

## 3. Discussion

The EHL is a unipennate complex of a muscle and tendon that arises from the middle half of the interosseous membrane and the fibula. The muscle belly lies between the extensor digitorum longus and the tibialis anterior muscle, becomes a tendon, and then passes under the superior and inferior extensor retinacula, to become inserted into the dorsobasal aspect of the distal phalanx of the hallux. The EHL is supplied by the anterior tibial artery and innervated by the deep peroneal nerve [12]. EHL contraction causes hallux extension, ankle dorsiflexion, and foot inversion.

Tendon rupture treatments vary by the timing of injury presentation. After acute injury, the rupture ends can be re-approximated without tension, rendering primary surgical repair possible using the Krakow or Kessler methods [5]. However, in patients with chronic injuries, the tendon contracture exceeds the maximal excursion, tension-free opposition of the ruptured ends is impossible, and secondary reconstructions, such as tendon transfer, autografting or allografting are required [13].

Tendon transfer to treat EHL ruptures usually employs the EDL of the second toe. After transfer of the second EDL to the distal end of the original EHL, the distal end of the second EDL is sutured to the third EDL to preserve extension of the second toe. This has a disadvantage, in which the biomechanical properties change due to the force directions of the original, and the second EDLs now vary [1]. Tendon autografts include the palmaris longus, semitendinosus, peroneus longus, and EDL tendons. However, tendon autografting may be painful and associated with functional donor site impairment. EHL reconstruction using a tensor fascia lata allograft has been reported. However, the risks include transplantation-associated hypersensitivity and viral transmission [14,15].

Scar tissue is a repair tissue that forms between the ruptured ends of tendons. Scars contain fibroblasts, fibrin, and collagen fibers [16]. During the first week of formation, scars are usually composed of blood cells, fibrin, and a few fibroblasts [8]. Commencing in week 2, granulation tissue and blood vessels form, and the fibroblasts produce connective tissue and collagen fibers. As the scar matures, type 3 collagen is converted into type 1 collagen (a component of intact tendons) and the collagen fibers become re-aligned along the direction of the tendon, improving the tensile strength [17]. Given these histological similarities to normal tendons, Yasuda et al. used scar tissue to successfully reconstruct neglected chronic ruptures of the Achilles tendon in 30 patients [10]. Yeo et al. reported good results when using scar tissue (for the first time) to reconstruct a chronic EDL rupture [11]. No tendon autograft or allograft is required, the incision is small, and the operative time is short [9,10,11].

In preoperative MRI, scar tissue exhibits a broad, fusiform-shaped high-level signal intensity or thin and heterogeneous high-level signal changes. Both types of tissue can be used for tendon reconstruction [10]. In our present case, the scar tissue was of the latter type. However, during surgery, we found that the thickness was similar to the normal tendon. If the running direction of the tendon does not match the MRI slice direction, tissue thickness may be inaccurate. Therefore, although scar tissue per se can be confirmed by MRI, the mechanical properties (thickness and tensile strength) should be evaluated during surgery, and then a decision can be made as to whether reconstruction is possible.

We are the first to use scar tissue to reconstruct a chronic EHL rupture. The postoperative AOFAS and FAAM scores were 90%, 95% (ADL subscale), and 94% (Sports subscale) respectively, comparable to those of a study on 20 patients with EHL ruptures [1]. There were some limitations to our work. First, we report on only one patient. Second, although the reconstructed tendon exhibited a homogenous signal intensity similar to a normal tendon in MRI performed 1 year after surgery, we did not confirm that the histological structures were identical. Bruns et al. reported that the scar tissue of transected Achilles tendons from sheep exhibited a near-normal tendon histology 12 months later. No long-term histological follow-up of human scar tissue has yet been reported [17]. Finally, this report is a 1-year follow-up result of ligament reconstruction using scar tissue and does not include long-term clinical outcomes. Yasuda et al. performed reconstruction using scar tissue in 30 chronic Achilles tendon rupture patients, and reported satisfactory clinical outcomes at the long-term follow-up of an average of 33 months (23–43 months) [10]. Further study is necessary to verify that the reconstructed EHL tissue can maintain long-term functions and keep patients asymptomatic.

We suggest that scar tissue reconstruction is possible not only for the Achilles and EHL tendons, but also other tendons. After additional research on the histological characteristics and chronology of scar tissue development, these reconstructions may serve as the primary treatment options for chronic tendon injuries. Autografting and allografting are associated with risks of donor site morbidity and infection.

## 4. Conclusions

We describe a rare reconstruction of a neglected EHL tendon rupture using interposed scar tissue. As midterm outcomes, the results were comparable to other treatments available for chronic ligament rupture, but no comparison was made at long-term follow-up. Since this procedure does not involve tendon autograft, there is no morbidity in the donor site and it has the advantages of small incisions and short operation time. After further study on the histological characteristics and chronology of scarring, these reconstructions may serve as the primary treatment options for chronic tendon injuries.

## Figures and Tables

**Figure 1 ijerph-18-12157-f001:**
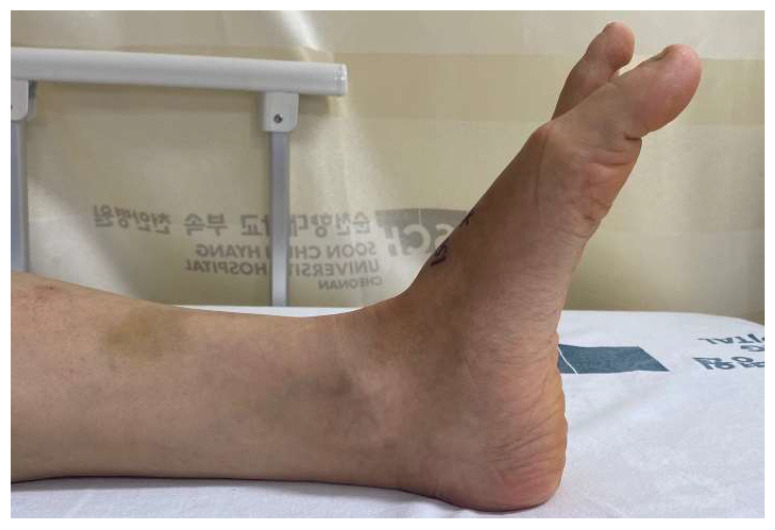
Preoperative lateral view of the right foot reveals loss of extension of great toe.

**Figure 2 ijerph-18-12157-f002:**
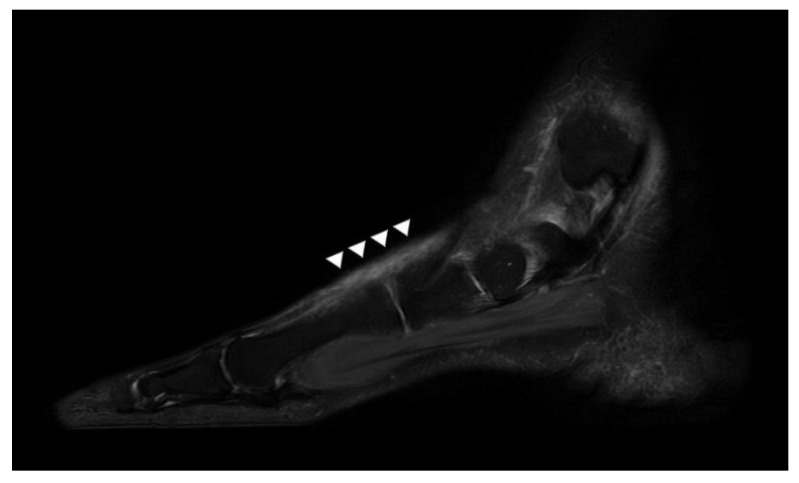
A preoperative-T2 weighted sagittal magnetic resonance image reveals that the EHL is discontinuous and that the intratendinous region is heterogeneous and of high signal intensity (white arrowhead).

**Figure 3 ijerph-18-12157-f003:**
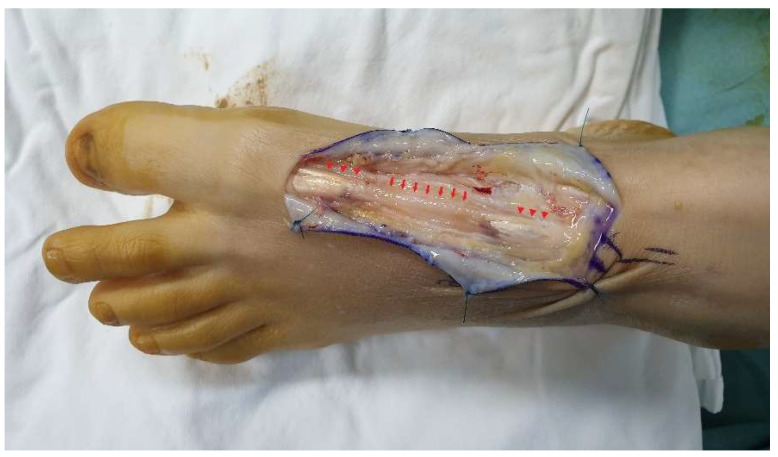
The gap between the tendon stumps (arrowhead) is filled with scar tissue (arrow).

**Figure 4 ijerph-18-12157-f004:**
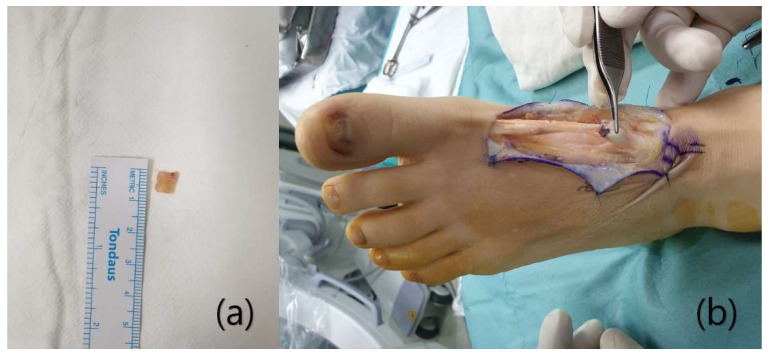
About 0.7 cm of the proximal scar tissue was resected to reduce the length of the EHL (**a**). The scar tissue was sutured to the proximal tendon stump (**b**).

**Figure 5 ijerph-18-12157-f005:**
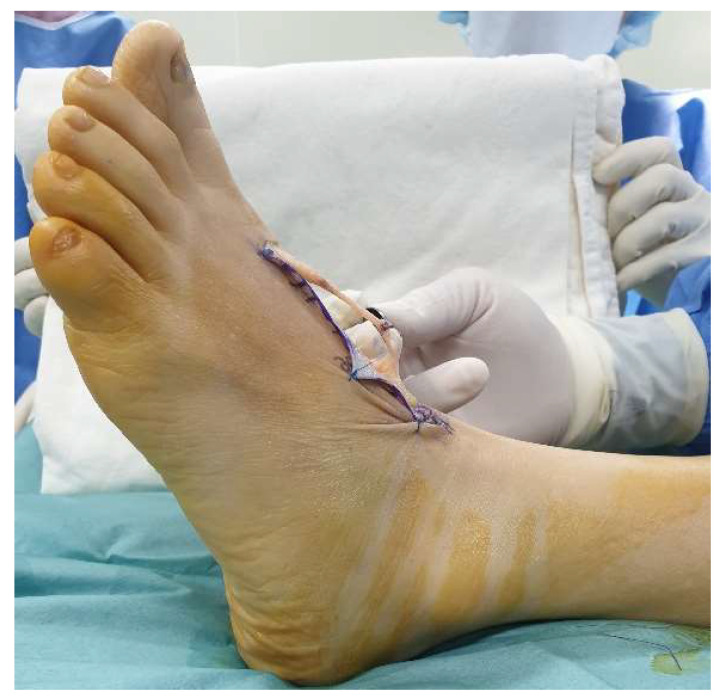
When the reconstructed tendon was pulled, the appropriate tension was maintained and the hallux was extended.

**Figure 6 ijerph-18-12157-f006:**
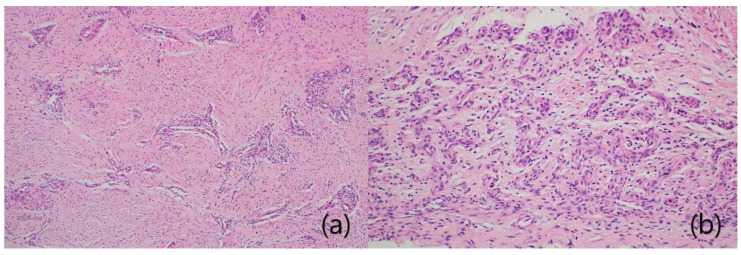
Microscopic views of cellular fibrous scar tissue with fibroblasts; vascular proliferation (**a**) and neovascularization (**b**) are evident [H&E stain, ×100 (**a**) and ×200 (**b**)].

**Figure 7 ijerph-18-12157-f007:**
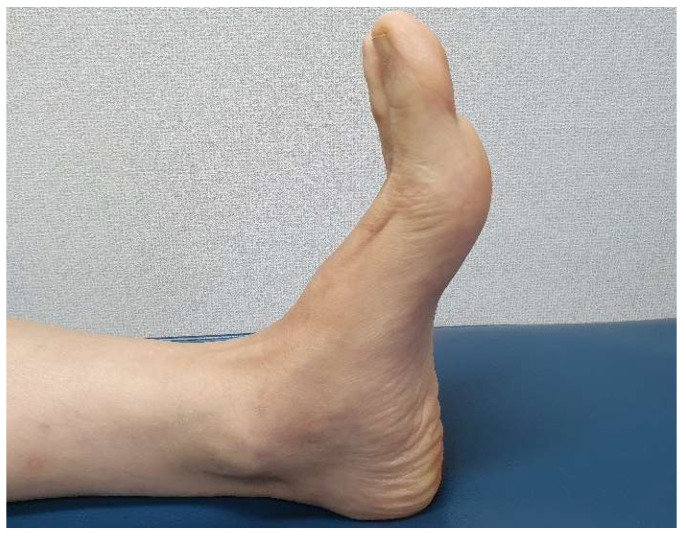
At the postoperative 6-month follow-up, the hallux extension strength was near-normal.

**Figure 8 ijerph-18-12157-f008:**
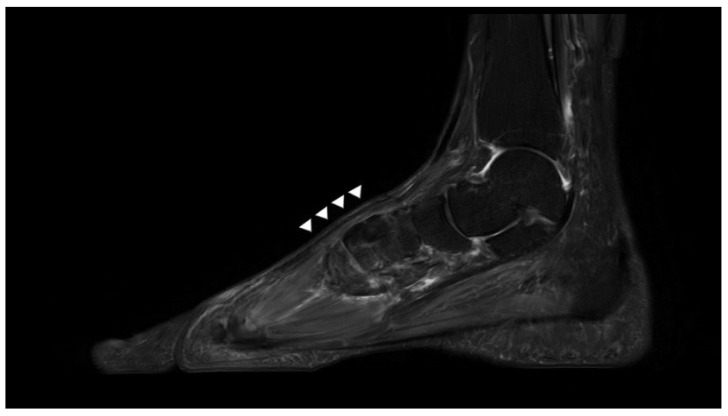
Postoperative T2-weighted MR image shows homogenous low-level signal intensity and continuity of reconstructed EHL tendon at 1 year postoperatively(white arrowheads).

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
