# Peer review of "Reconstruction of a Neglected, Extensor Hallucis Longus Tendon Rupture Using Interposed Scar Tissue: A Case Report and Literature Review"

_ijerph, 2021, doi:10.3390/ijerph182212157_

Round 1

Reviewer 1 Report

Comments to the Authors of manuscript number: ijerph-1425520 entitled Reconstruction of a neglected, extensor hallucis longus tendon rupture using interposed scar tissue: A case report and literature review.

Authors investigated the extensor hallucis longus effect reconstruction previously injured.

The study has been performed in proper manner. Each part of the manuscript is well presented and written.

I recommend minnor grammar revision.

0.- Abstract. Is well structured and describe the research.

1.- Introduction. Nothing remarkable

2.- Case presentation

Have authors analyzed other ligaments, joints and muscles foot in the MRI before and after reconstruction?

Could others foot structures have been damaged prior to reconstruction due to biomechanical failure of the EHL? Did authors analyzed them?

In Figure 8 a talar dome synovitis is shown. How have influenced the EHL reconstruction in other foot structures?

3.- Discussion. Ok

4.- Conclusions. Ok.

Author Response

Response to Reviewer 1 Comments

First of all, the authors would like to thank the reviewer for the excellent advices.

Point 1: Have authors analyzed other ligaments, joints and muscles foot in the MRI before and after reconstruction?

Response 1: Thanks for the good point. There were no abnormal findings other than EHL rupture on the preoperative MRI, and no other ligament and muscle problems were found on the MRI at 1 year postoperatively.  The above has been added to the text. Thank you.

Point 2: Could others foot structures have been damaged prior to reconstruction due to biomechanical failure of the EHL? Did authors analyzed them?

 Response 2: On the physical examination performed before surgery, no biomechanical dysfunction was observed in any ligaments other than EHL, and there were no abnormalities in other joints of the foot. Also, there were no other anatomical abnormalities on the MRI. Accordingly, the authors concluded that the EHL rupture did not affect other structures. The above has been added to the text.

Point 3: In Figure 8 a talar dome synovitis is shown. How have influenced the EHL reconstruction in other foot structures?

 Response 3: Figure 8 is a T2 weight MRI taken one year after surgery. At the one year follow up, the patient did not complain of pain or swelling in the ankle joint, and there was no tenderness or ROM pain on physical examination. Although the high signal intensity of the talar dome was observed as the reviewer pointed out, the authors judged it to be a simple joint fluid collection based on the patient's asymptomatic condition. Thanks for the sharp point.

Reviewer 2 Report

The authors present a case report of a surgical repair of chronic tendon ruptures (extensor hallucis longus tendon) using interposed scar tissue, already described in other locations such as Achilles tendon and the extensor digitorum longus tendon. Although, it seems to be the first to see that the use of scar tissue to reconstruct a chronic rupture of extensor hallucis longus tendon is reported. The originality of the work is more for the location of the lesion than for the novelty of the technique, since in the manuscript itself a work from 2007 is cited (ref 9).

The work is correctly exposed and details those aspects necessary for the follow-up of the case.

The main limitation of the work is that, because the histological structures were not identical at 6 months, it is necessary to carry out a greater long-term follow-up to verify that the tissue continues to respond to the functional needs of the foot, keeping the patient asymptomatic. Therefore, I consider it necessary to cite other works, if there are, that deal with the chronology of the development of scar tissue within a period of more than 1 year in humans.

It would be interesting to specify that the concluding observations on the results obtained have been given in the short or medium term (1 year).

The authors indicate that " Donor site morbidity and infection are obviously impossible, the incision is small, and the operative time is short.", however, making this claim is health sciences is risky, as zero risk is unlikely to exist. I think this phrase should be reconsidered.

Author Response

Response to Reviewer 2 Comments

First of all, the authors would like to thank the reviewer for the excellent advices.

Point 1: The main limitation of the work is that, because the histological structures were not identical at 6 months, it is necessary to carry out a greater long-term follow-up to verify that the tissue continues to respond to the functional needs of the foot, keeping the patient asymptomatic. Therefore, I consider it necessary to cite other works, if there are, that deal with the chronology of the development of scar tissue within a period of more than 1 year in humans.

Response 1: Thanks for the good point. As pointed out, this report is a one-year follow-up result of ligament reconstruction using scar tissue. The main limitation of the study is that it does not include clinical results following long-term follow-up. In this regard, we mentioned the paper on the long-term follow-up results after reconstruction using scar tissue in Achilles chronic rupture. (There have been no long-term follow-up studies of other ligaments.) In addition, it would be good to find that histologically identical or similar to the original ligament through a biopsy after reconstruction, but this seems to violate research ethics. This is the reason why the authors did not recommend biopsy to patients at 1 year postoperative follow-up. It would be grateful if you could understand this situation.

Point 2: It would be interesting to specify that the concluding observations on the results obtained have been given in the short or medium term (1 year).

 Response 2: Thanks for the good advice. As pointed out, it was clarified that this report is for mid-term results, and it was added to the conclusion that no comparison was made with long-term follow-up.

Point 3: The authors indicate that " Donor site morbidity and infection are obviously impossible, the incision is small, and the operative time is short.", however, making this claim is health sciences is risky, as zero risk is unlikely to exist. I think this phrase should be reconsidered.

 Response 3: We totally agree with the reviewer's comments. Zero risk is unlikely to exist in health sciences. The authors have corrected the phrase. Thanks for the good point.
